# TCI: Mitigating Hallucination in LVLMs via Text Contrastive Intervention

## Abstract

Large Vision-Language Models (LVLMs) have achieved remarkable progress across a wide range of tasks by integrating visual and textual information. Yet they still suffer from a common issue: hallucination, where the generated text fails to accurately align with visual inputs. Existing contrastive methods primarily intervene on the visual modality, perturbing images to indirectly amplify language priors, but fail to directly target text to expose and mitigate text bias. To address this, we propose **T**ext **C**ontrastive **I**ntervention (TCI), a training-free approach that amplifies visual information in those attention layers most susceptible to language bias. Our method is inspired by a key observation: the *repetition phenomenon*, where LVLMs tend to verbatim repeat text when conflicts arise between the images and accompanying text. We hypothesize this behavior stems from language priors—a critical cause of hallucinations. TCI operates in two steps: first quantifying per-layer attention shifts under text perturbation to identify the layers where visual attention is most compromised; then we selectively boost the corresponding visual-attention weights during generation, steering the model away from text bias. Extensive experiments demonstrate that TCI significantly reduces hallucinations while requiring only a small amount of data, demonstrating its effectiveness and efficiency.

## 1 Introduction

Building on Large Language Models (LLMs) (Vaswani et al., 2017; Zheng et al., 2023; Bai et al., 2023a), Large Vision-Language Models (LVLMs) have integrated visual and linguistic modalities, demonstrating remarkable potential in real-world tasks such as image captioning and visual question answering (VQA) (Li et al., 2022; 2023a; Zhu et al., 2024; Liu et al., 2023; Bai et al., 2023b; Liu et al., 2024b). However, they are plagued by a critical issue: hallucinations (Liu et al., 2024a), where generated content misaligns with visual inputs (e.g., falsely asserting the presence of non-existent objects). This undermines the reliability of LVLMs in practical applications, with severe implications for high-stakes domains like autonomous driving (Chen et al., 2023a) and medical diagnosis (Hu et al., 2023).

Prior research has identified two primary sources of hallucinations:1) Models. Visual encoders, such as CLIP Radford et al. (2021)) may inaccurately capture visual features, leading to errors in object recognition or attribute judgment (Rohrbach et al., 2018). Moreover, since LLMs constitute the majority of parameters in LVLMs, the models tend to prioritize linguistic knowledge patterns, causing over-reliance on language priors (Guan et al., 2024; Leng et al., 2024; Rohrbach et al., 2018). 2) Data. Noisy annotations (e.g., misalignment between text and images) (Yu et al., 2024a; Yue et al., 2024) and statistical biases (e.g., frequent object co-occurrences) (Li et al., 2023b; Rohrbach et al., 2018; Zhou et al., 2023; Schrodi et al., 2025) in training corpora further exacerbate hallucinations.

Existing methods to mitigate hallucinations fall into two categories. One approach is to use Supervised Fine-Tuning (Chen et al., 2023b; Yue et al., 2024) or Reinforcement Learning (Yang et al., 2025b; Yu et al., 2024b; Kumar et al., 2025; Xing et al., 2025), but these require extensive manual annotations and computational resources. Alternatively, image-based contrastive decoding methods operate in two stages: as shown in Figure 1, they first amplify hallucinations by corrupting input images, and then mitigate such hallucinations by contrasting shifts in model internal states and output distributions before and after corruption. (Leng et al., 2024; Chen et al., 2025; An et al., 2024; He

et al., 2025). However, such methods amplify language priors indirectly rather than targeting them directly.

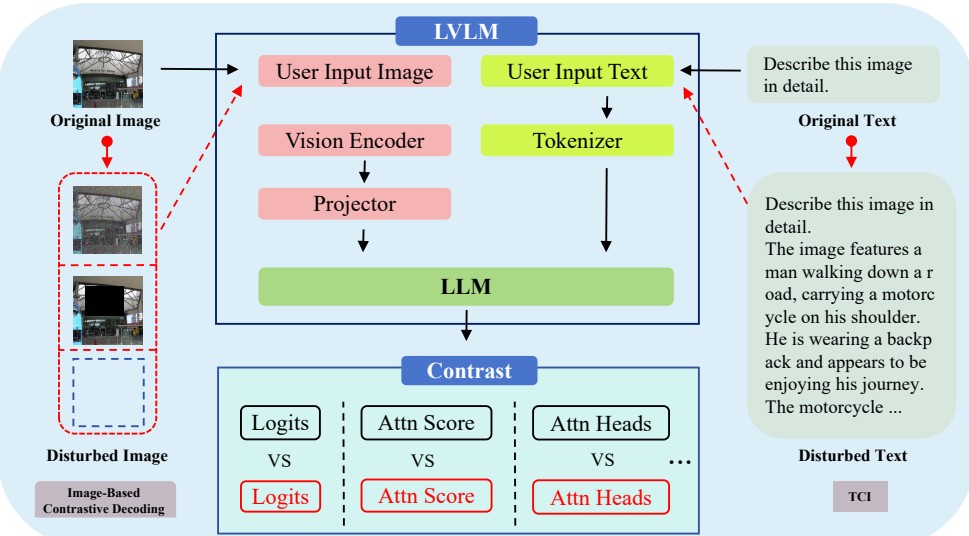

Figure 1: Comparison between Image-based Contrastive Decoding and our proposed TCI Method. (Left) Image-based methods perturb the visual input—via global noise, local mask, or complete removal—to amplify hallucinations, then mitigate hallucinations by contrasting differences in probability distributions, attention scores, or attention heads before and after perturbation. (Right) In contrast, our TCI method perturbs the text modality to directly leverage language prior for hallucination amplification, measures layer-wise attention shifts, and selectively intervenes on the most text-biased layers to reinforce visual grounding.

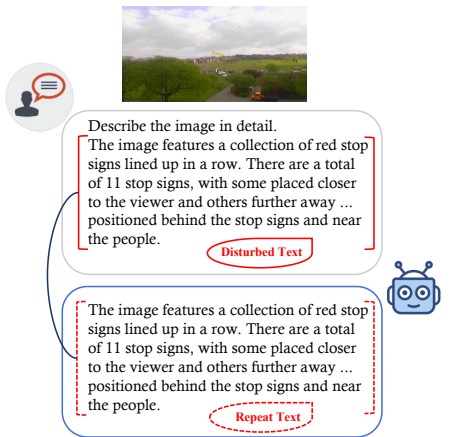

Figure 2: An example of *repetition phenomenon*. Perturbing the input caption makes the model repeat it verbatim, despite visual contradictions.

**Repetition phenomenon.** To address this limitation, we explore direct amplification of language priors by perturbing input text and observe a *repetition phenomenon*: LVLMs tend to repeat input text even when it conflicts with the image. Figure 2 illustrates an example of this phenomenon. We hypothesize that the emergence of this behavior and the occurrence of hallucinations share the same cause: the model tends to neglect genuine visual information during the generation process.

We validate this by randomly sampling 1,000 images from the COCO 2014 validation set (Lin et al., 2014), apply random caption perturbations, and compute the average attention weight across all layers for both LLaVA-1.5-7B (Liu et al., 2024b) and Qwen-VL-Chat-7B (Bai et al., 2023b). The experimental details can be found in the Appendix C. As shown in Figure 3 and Table 1, perturbation triggers a marked drop in attention to image regions and a compensatory rise in text attention, confirming that weakened visual engagement drives hallucination.

**Text Contrastive Intervention**. Leveraging repetition phenomenon, we propose **T**ext **C**ontrastive **I**ntervention (TCI), a training-free intervention method applied during the forward pass to mitigate hallucinations. As shown in Figure 1, unlike image-based contrastive methods, TCI perturbs text, contrasts attention shifts across layers, and enhances visual attention in selected layers during the forward pass to reduce text bias. TCI precisely targets layers prone to text bias, avoiding unnecessary adjustments to visually sensitive layers. Our analysis reveals that shallow and middle attention layers

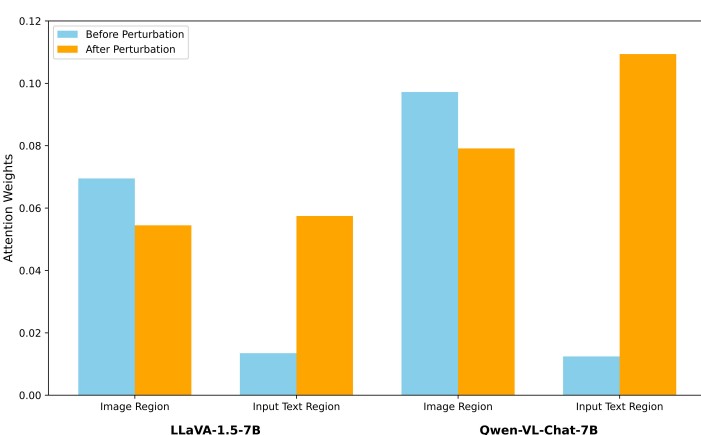

Figure 3: Average attention weights on visual and text regions with/without text perturbation. In both models, text perturbation causes a clear decrease in attention to image regions and a compensatory increase in attention to the (misleading) text regions.

Table 1: Hallucination metrics with/without text perturbation. We evaluate hallucination levels using the CHAIR metric. Results show a significant increase in hallucinations after text perturbation (higher $C_I$ and $C_S$ values indicate more hallucinations).

| Method | LLaVA-1.5-7B | | Qwen-VL-Chat-7B | |
|---|---|---|---|---|
| | $C_S \downarrow$ | $C_I \downarrow$ | $C_S \downarrow$ | $C_I \downarrow$ |
| w/o text perturbation | 50.00 | 13.40 | 48.70 | 13.20 |
| w/ text perturbation | 61.10 | 32.90 | 60.80 | 31.00 |

are more likely to neglect visual information after perturbation, indicating their key role in cross-modal fusion.

Experiments on established LVLM hallucination benchmarks validate TCI's superiority over existing decoding strategies, demonstrating its effectiveness and efficiency in alleviating hallucinations. On the POPE benchmark, TCI improves average accuracy by 7.5% for LLaVA-1.5-7B and 4.6% for Qwen-VL-Chat-7B, with corresponding F1-score gains of 9.7% and 5.0%. On CHAIR, LLaVA-1.5-7B with TCI reduces hallucination metrics by 43.8% and 41.4%, while Qwen-VL-Chat-7B achieves reductions of 5.6% and 9.1%.

**Contributions.** Our contributions are summarized as follows:

- Unlike existing image-corruption-based contrastive decoding methods, we amplify hallucinations directly by perturbing text to exploit language priors, approaching the problem from a linguistic modality perspective.

- We propose **T**ext **C**ontrastive **I**ntervention (TCI), a training-free method that mitigates hallucinations in LVLMs by enhancing visual attention during the forward pass, thereby reducing over-reliance on language priors.

- Extensive experiments on LVLMs demonstrate that TCI significantly improves performance across multiple widely adopted hallucination benchmarks.

## 2 RELATED WORK

### 2.1 MITIGATING HALLUCINATIONS IN LVLMS

In contrast to Large Language Models, hallucinations in LVLMs refer to a mismatch between the generated text and the content of the input image (Liu et al., 2024a). Such hallucinations typi-

cally stem from strong language priors or from imbalanced vision–language training data. Existing mitigation techniques can be categorized into two groups: post-training methods and training-free interventions.

Post-training methods employ supervised finetuning (Chen et al., 2023b; Yue et al., 2024), or reinforcement learning based schemes to improve cross-modal alignment Yang et al. (2025b); Yu et al. (2024b); Kumar et al. (2025); Xing et al. (2025). While effective, these approaches require additional data and computational resources.

Training-free interventions instead modify the decoding process during inference, intervening at the level of output probabilities (Leng et al., 2024; Wang et al., 2024; Huo et al., 2025), attention networks (Yin et al., 2025; He et al., 2025; Zhang et al., 2024), or feed-forward networks (Yang et al., 2025a). A prominent subclass of these is Contrastive Decoding: one first generates outputs under an intentionally *inconsistent* text–image pairing to amplify hallucinations, then compares them to outputs under a *consistent* pairing to derive corrective signals for decoding. To date, inconsistency has been introduced chiefly by perturbing the image or by inserting disturbance instructions into the text input (Leng et al., 2024; Huang et al., 2024; Chen et al., 2025; Huo et al., 2025; An et al., 2024; He et al., 2025; Wang et al., 2024). However, these methods do not explicitly exploit the model's underlying language priors.

In comparison, our approach first amplifies linguistic bias by fully replacing the input text, then identify and enhance specific attention layers that are more image-aware during inference,thereby mitigating hallucinations.

## 2.2 Text Bias in LVLMs

Large Vision–Language Models (LVLMs) are typically composed of a visual encoder, a projection module, and a pretrained language model. Given that the language model often contains far more parameters than the vision encoder, LVLMs inherently inherit strong linguistic priors—or "language bias" (Wu et al., 2022; Han et al., 2022; Ghosh et al., 2025; Zhu et al., 2020; Lee et al., 2025). Text bias, in turn, can be viewed as one manifestation of this phenomenon. Deng et al. (2025) demonstrates that perturbing input text causes models to over-rely on textual information, resulting in degraded performance. Similarly, Hua et al. (2025) finds that when image–text pairs conflict, overall accuracy declines, and with specific "promotion" heads consistently amplifying either text or image information. However, these studies focus on general performance under conflict rather than the hallucination problem.

More recently, Liu et al. (2025); He et al. (2025) identifies that LVLMs generate identical hallucinations whether an image is provided or not, and mitigates such errors by intervening on attention heads during decoding. This observation and the repetition phenomenon we discovered both indicate the tend towards text bias.

## 3 Method

In this section, we first compare the shifts in attention to visual information at the layer level, before and after text perturbation (Section 3.1). Based on the magnitude of these shifts, we then intervene in layers exhibiting the most significant deviations, enhancing their attention to visual information during the model's forward pass phase (Section 3.2).

### 3.1 Identify Text-Aware Layers

To identify the attention layers that are more sensitive to text information during generation, we compare the attention shift patterns of each layer during the forward pass, before and after the input text is perturbed.

As showed in Figure 4(a), given an LVLM parameterized by $\theta$, we construct inputs for both the pre- and post-text perturbation scenarios. We first define a set of original inputs $\{(v_i, x_0)\}_{i=1}^{B}$, which contains $B$ image-text pairs. Here, $x_0$ is fixed as "Describe the image in detail.", $v_i$ represents the $i$-th input image, and the corresponding output is denoted as $g_i$:

$$g_i = \text{LVLM}_\theta(v_i, x_0) \tag{1}$$

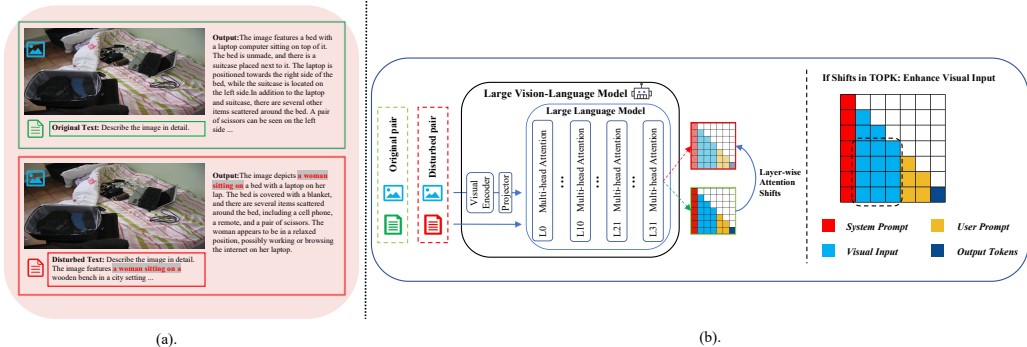

Figure 4: (a) Example image–text pairs before and after perturbation: we randomly append a caption generated for a different image to serve as the perturbation text. (b) Pipeline of our TCI method: The image-text pairs are fed into the model, and layer-wise attention shifts toward visual regions are computed and compared during generation. After ranking these shifts, the top-k layers with the largest shifts have their attention to visual regions enhanced during the forward pass, suppressing over-reliance on language priors.

Subsequently, we construct a set of perturbed texts. For each $v_i$, we randomly splice the description $g_j$ of a different image with $x_0$ to form the perturbed text $x'_i$ (where $i \neq j$ and $i, j \in (0, B)$):

$$x'_i = x_0 + \text{random}(g_j), \ \{ \ i, j \mid i, j \in (0, B), \ i \neq j \} \tag{2}$$

We then obtain the perturbed output $g'_i$:

$$g'_i = \text{LVLM}_\theta(v_i, x'_i) \tag{3}$$

As illustrated in Figure 4(b), during the generation phase, there are four types of inputs: System Prompt, Visual Input, User Prompt, and generated Tokens, denoted as $S, V, X$, and $G$, respectively. For the input $A = \text{concat}(S, V, X, G)$, we compute multi-head self-attention for the $l$-th layer of the model:

$$A^l = A^{l-1} + \text{MultiHead}^l(A^{l-1}) \tag{4}$$

For the $i$-th text-image pair before perturbation, when generating the $g$-th token, the attention to the $j$-th token in the image region is $A^l_{(g,j),i}$. Similarly, for the $i$-th text-image pair after perturbation, we have $A^{l'}_{(g,j),i}$.

For all inputs, we calculate the attention shift of the $l$-th layer averaged over all image regions before and after perturbation:

$$AS^l = \frac{1}{B} \cdot \frac{1}{V} \sum_{i=0}^{B} \sum_{j=0}^{V} \left( A^l_{(g,j),i} - A^{l'}_{(g,j),i} \right) \tag{5}$$

V denotes the number of visual tokens (e.g., 576 for LLaVA-1.5-7B features), ensuring we average shifts across all visual regions to avoid bias from individual token outliers. The double summation over i (samples) and j (visual tokens) further stabilizes the metric, reducing variance from single samples.

Finally, we sort all $AS^l$ values in descending order. Layers with larger $AS^l$ values are more sensitive to text, thus are more susceptible to the influence of language priors, and are more likely to overlook visual information when perturbed.

## 3.2 ENHANCE VISION IN FORWARD PASS

After identifying the top text-aware layers, during the generation phase of the model, we enhance their attention to visual regions. This forces the attention layers to reduce text bias and focus on vision regions. Specifically, when the $l - th$ layer generates the $g - th$ token in an autoregressive manner, the attention to various regions is denoted as $A^l_{(S,U,V,g-1)\leftarrow g}$. We only enhance the attention to visual information, while keeping the attention to other regions unchanged:

$$\overline{A}^l_{g\leftarrow(S,U,V,g-1)} = (1 + \alpha) \cdot A^l_{g\leftarrow(:,:,V,:)}, \qquad \text{when } g > 0 \tag{6}$$

The enhanced attention $\overline{A}^l$ is then integrated into the layer-wise feature propagation. First, the multi-head attention outputs are concatenated and projected:

$$\text{MHA}^l_{\text{enhanced}} = \text{Concat}\left(\text{head}_1(\overline{A}^l), \dots, \text{head}_h(\overline{A}^l)\right) \cdot W^O \tag{7}$$

$W^O$ is the projection matrix for the output of multi-head attention. Finally, the probability of generating the next token g is obtained as:

$$p_\theta(g \mid S, U, V, G_{<g}) = \text{Softmax}\left[\text{logit}\theta(g \mid S, U, V, G< g)\right] \tag{8}$$

$G_{<g}$ represents generated tokens before position g (i.e., $[g_1, g_2, ..., g_{g-1}]$). The model selects the next token $g$ based on the configured decoding strategy and probability distribution. The two-stage procedure is shown in Appendix C.4.

The enhancement factor $\alpha$ is a critical hyperparameter that balances visual attention amplification and generation stability. Through preliminary experiments, we observe that when $\alpha > 0$, it can mitigate text bias and reduce hallucinations.

However, if $\alpha$ is excessively large, the model may over-focus on visual information, which disrupts its language capabilities, leading to anomalies such as repetitive sentence generation. Conversely, when $\alpha < 0$, the model tends to underutilize visual information, resulting in performance degradation.

Moreover, different models and tasks exhibit varying sensitivities to $\alpha$. We provide further ablation studies and analysis in Section 4 to validate its effectiveness and robustness.

## 4 EXPERIMENTS

### 4.1 EXPERIMENTAL SETUP

**Datasets and Metrics**

**POPE.** The Polling-based Object Probing Evaluation(POPE) (Li et al., 2023b) is designed to detect object hallucinations. It adopts a fixed yes-or-no question format: "Is there a <object> in the image?" to evaluate the model's ability to determine the presence of specific objects in given images. Images are sourced from COCO (Lin et al., 2014). Based on object sampling strategies, POPE is divided into three subsets: random, popular, and adversarial. Each subset contains 3,000 questions, with answers balanced equally between "yes" and "no". Evaluation metrics include Accuracy, Precision, Recall, and F1 score.

**CHAIR.** The Caption Hallucination Assessment with Image Relevance (CHAIR) (Rohrbach et al., 2018) is an effective metric for evaluating object hallucinations in image captioning tasks. Specifically, it identifies hallucinations by comparing whether objects in generated captions exist in the ground truth set. CHAIR evaluates two levels: $C_I$ and $C_S$, stand for object-level and caption-level,respectively:

$$C_I = \frac{|\{\text{hallucinated objects}\}|}{|\{\text{all mentioned objects}\}|} \tag{9}$$

$$C_S = \frac{|\{\text{captions w/hallucinated objects}\}|}{|\{\text{all captions}\}|} \tag{10}$$

Table 2: **Average accuracy and F1 scores of POPE**, with best in **bold** and second-best underlined. Results of popular, adversarial, and random splits are shown in Table 8 in Appendix D.2.

| Method | LLaVA-1.5-7B | | Qwen-VL-Chat-7B | |
|---|---|---|---|---|
| | Accuracy | F1 Score | Accuracy | F1 Score |
| Direct Sample | 81.38 | 79.65 | 83.59 | 81.70 |
| Greedy | 85.19 | 86.10 | 86.81 | 85.84 |
| VCD | 83.70 | 84.82 | 85.26 | 84.35 |
| OPERA | 85.69 | 85.60 | 86.99 | 85.99 |
| **TCI** | **87.52** | **87.39** | **87.46** | **86.67** |

Table 3: **CHAIR results**, with best in **bold** and second-best underlined.

| Method | LLaVA-1.5-7B | | Qwen-VL-Chat-7B | |
|---|---|---|---|---|
| | $C_S \downarrow$ | $C_I \downarrow$ | $C_S \downarrow$ | $C_I \downarrow$ |
| Direct Sample | 56.72 | 17.40 | 48.16 | 13.70 |
| Greedy | 49.88 | 14.28 | 45.00 | 12.16 |
| VCD | 52.24 | 15.32 | 46.92 | 12.98 |
| OPERA | 44.6 | 12.80 | - | - |
| **TCI** | **31.88** | **10.20** | **42.44** | **11.92** |

We randomly sample 500 images from the COCO 2014 validation set and repeat experiments five times with different random seeds. For all LVLMs, the input prompt is unified as "Describe this image in detail." to generate descriptions. We report average results for each metric.

**LLaVA-Bench.** LLaVA-Bench (In-the-Wild) is a comprehensive evaluation dataset consisting of 24 images with 60 open-ended questions, specifically designed to evaluate an LVLM's performance on challenging tasks and its generalization to novel domains. Following prior work (Huang et al., 2024; He et al., 2025), we leverage GPT-4o as an automatic judge to score model responses across three dimensions: accuracy, informativeness (level of detail), and naturalness. The prompt of GPT-4o is shown in Table 7 in Appendix C.5.

**Models.** We validate our method on two representative models: LLaVA-1.5-7B (Liu et al., 2024b) and Qwen-VL-Chat-7B (Bai et al., 2023b). The key difference lies in their projection mechanisms: LLaVA-1.5-7B employs an MLP projector, while Qwen-VL-Chat-7B uses cross-attention.

**Baselines.** We compare our method with two baselines: VCD (Leng et al., 2024) and OPERA (Huang et al., 2024). VCD amplifies hallucinations by perturbing images with Gaussian noise, then intervenes by comparing token probability changes before and after perturbation. OPERA penalizes hallucinatory candidates via beam search during decoding and performs backtracking.

**Implementation Details. Stage 1:** We randomly select 1,000 images from the COCO 2014 validation set and generate original outputs using the prompt "Describe this image in detail". For each image, perturbed texts are randomly sampled from these original outputs. **Stage 2:** The top-k parameter is fixed at 5 for both models. For LLaVA-1.5-7B, $\alpha = 4$ (POPE, CHAIR), $\alpha = 2$ (LLaVA-Bench); for Qwen-VL-Chat-7B, $\alpha = 1.7$ (POPE), $\alpha = 2.5$ (LLaVA-Bench) and $\alpha = 2.8$ (CHAIR). Decoding adopts greedy search with a maximum token length of 512. More implementation details please refer to Appendix C.

## 4.2 EXPERIMENTAL RESULTS

**Layer Susceptibility to Text Perturbations.** In the Figure 7 of Appendix D.1, we visualize the layer-wise shifts in attention toward visual regions before and after text perturbation. The two models exhibit distinct shift patterns : the top 5 layers with the largest shifts for LLaVA-1.5-7B are $\{0, 1, 14, 15, 17\}$, whereas for Qwen-VL-Chat-7B they are $\{0, 1, 17, 20, 31\}$. Notably, these layers lie predominantly in the shallow to middle part of the network, indicating that early and intermediate layers are most prone to neglecting visual information, relying more heavily on language priors and contributing to hallucinations.

Table 4: **Evaluation results of LLaVA-Bench (In-the-Wild)**, metrics are scored by GPT-4o on a scale of 10.

|  | Accuracy | Detailedness | Naturalness |
|---|---|---|---|
| LLaVA-1.5 | 5.133 | 5.483 | 7.050 |
| w/TCI | **5.367** | **5.683** | **7.117** |
| Qwen-VL | 6.033 | **6.200** | 7.167 |
| w/TCI | **6.476** | 5.850 | **7.550** |

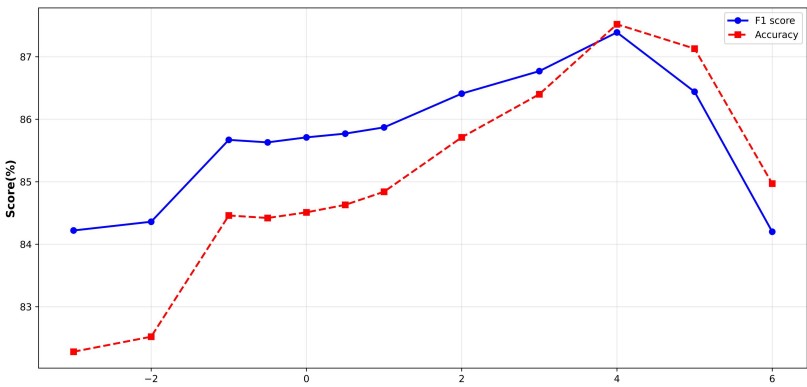

Figure 5: **Ablation of** $\alpha$. Results on POPE dataset with LLaVA-1.5-7B.

**Results on POPE.** Table 2 presents average performance for LLaVA-1.5-7B and Qwen-VL-Chat-7B across the three POPE subsets. We observe that, in terms of average Accuracy, the TCI method improves LLaVA-1.5-7B and Qwen-VL-Chat-7B by 7.5% and 4.6%, respectively. For average F1 Score, the improvements are 9.7% and 5.0%, respectively. Additionally, our method consistently outperforms previous approaches on multiple subsets, demonstrating its effectiveness as a training-free strategy across different performance levels. This improvement can be attributed to TCI's dual role: it not only emphasizes useful visual information but also suppresses language priors. The specific results of the three splits are shown in Appendix D.2.

**Results on CHAIR.** As shown in Table 3, our method significantly outperforms all baseline methods in the image captioning task. Specifically, for LLaVA-1.5-7B with TCI, $C_S$ and $C_I$ decrease by 43.8% and 41.4%, respectively; for Qwen-VL-Chat-7B, the corresponding reductions are 5.6% and 9.1%. These results confirm TCI's ability to mitigate hallucinations by rebalancing attention toward authentic visual content during generation.

**Results on LLaVA-Bench.** Table 4 presents the GPT-4o evaluation on LLaVA-Bench (In-the-Wild). The results demonstrate that TCI improves model accuracy and effectively mitigates hallucinations in the generated captions, while preserving comparable levels of detailedness and naturalness.The effectiveness of TCI can be further illustrated by additional cases, as shown in Figure 8 and 9 in Appendix E.

### 4.3 ABLATION STUDY

**Impact of Hyperparameter on Performance.** To assess how the enhancement factor $\alpha$ affects model performance, we sweep $\alpha$ from -3 to 6 on LLaVA-1.5-7B using the POPE benchmark. As shown in Figure 5, When $\alpha > 0$, hallucinations are effectively mitigated, confirming the utility of visual attention amplification. Excessively large $\alpha$ degrades performance, as the model over-focuses on irrelevant visual details during inference. When $\alpha < 0$, visual attention is suppressed, forcing the model to rely more on language priors and thus amplifying hallucinations.

**Impact of Attention-Shift-Guided Layer Selection**. To validate the role of attention shift ($AS^l$), we randomly selected 5 attention layers in LLaVA-1.5-7B for enhancement (with $\alpha = 4$). Experi-

Table 5: **Ablation of random 5 layers.** Results on POPE Adversarial with LLaVA-1.5-7B.We test random 5 layers for 3 times.

| Method | Accuracy | F1 score |
|---|---|---|
| Greedy | 79.77 | 81.78 |
| w/TCI Top 5 Layers | 83.77 | 84.10 |
| w/TCI Random 5 Layers | 79.99 | 82.03 |

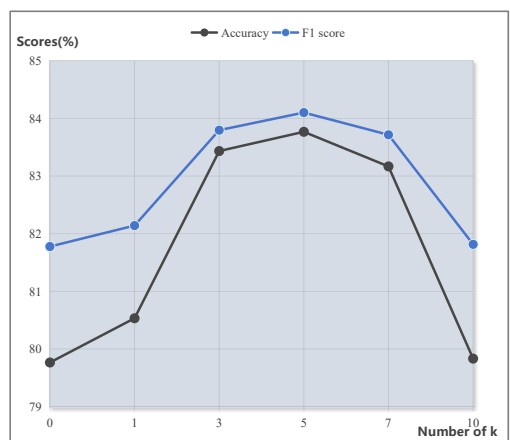

Figure 6: **Ablation of $k$.** Results on POPE Adversarial with LLaVA-1.5-7B.

ments were repeated three times with different random seeds, and results were averaged. As shown in Table 5, enhancing randomly selected layers yields limited improvement in the POPE adversarial subset. This indicates that interventions targeting layers with larger attention shifts are more effective in mitigating hallucinations.

**Impact of Number of Enhanced Layers** ($k$). For the selection of top-$k$ layers, we vary the number of top-$k$ attention-shift layers selected for enhancement, with $k \in \{1, 3, 5, 7, 10\}$, and measure Accuracy and F1 on POPE using LLaVA-1.5-7B. Results in Figure 6 show a non-monotonic trend: increasing $k$ initially boosts hallucination mitigation, reaches an optimum at $k = 5$, then declines as additional layers introduce excess visual noise or dilute the effect. These ablations validate our choice of $k = 5$ for a balanced trade-off between effectiveness and overcorrection.

## 5 CONCLUSION

In this paper, we first identify a *repetition phenomenon*: LVLMs tend to repeat input text even when it conflicts with the image content. This phenomenon indicates that hallucinations arise from text bias and the neglect of visual information. Building on this insight, we propose TCI (Text-Contrastive Intervention), a training-free method that guides the model to prioritize visual information over language priors via interventions during the forward pass. Our analysis reveals that shallow and middle layers play a more critical role in cross-modal information fusion; thus, TCI specifically targets these layers for intervention. Extensive experiments demonstrate that our method consistently outperforms baselines in reducing hallucinations across various LVLMs and evaluation metrics, validating its effectiveness and generality.

## 6 REPRODUCIBILITY STATEMENT

The datasets and models we used in the experiment are both open-source. We have provided experimental codes for the LLaVA-1.5-7B model in the supplementary materials. We will release the code

for the Qwen model and the required experimental environment if the paper is accepted to ensure its reproducibility.

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

## A    LIMITATIONS

Despite its simplicity and effectiveness, TCI has certain limitations. Our intervention focuses exclusively on attention layers, without addressing other architectural components—such as individual attention heads or feed-forward networks(FFNs)—that may also contribute to hallucination. Investigating and intervening on these components remains an important direction for future work.

## B    DETAILS OF USING LARGE LANGUAGE MODELS

We use the Large Language Models for proofreading when writing, specifically, we use it in various chapters of the article to improve readability and professionalism, such as modifying word order, replacing certain words, etc.

## C    IMPLEMENTATION DETAILS

### C.1    EXPERIMENTAL SETUP

All LLaVA-1.5-7B experiments were conducted on an NVIDIA RTX 4090 GPU, while Qwen-VL-Chat-7B evaluations ran on an NVIDIA H20 GPU.

Table 6: Algorithm 1: Text Contrastive Intervention(TCI)

---

**Algorithm 1: Text Contrastive Intervention(TCI)**

---

**Input:** LVLM$_\theta$, batch of image-text pairs $\mathcal{D} = \{(v_i, x_0)\}_{i=1}^B$ (where $x_0$ = "Describe the image in detail."), enhancement factor $\alpha$

---

**Stage 1: Identify Text-Aware Layers**
1: **For** each $(v_i, x_0) \in \mathcal{D}$ **do**
2:   Get original output                            ▷ Equation (1)
3:   Create perturbed text                          ▷ Equation (2)
4:   Get perturbed output                           ▷ Equation (3)
5:   **For** each layer $l$ **do**
6:     Extract layer-wise attention
7:     Compute attention shifts                     ▷ Equation (5)
8: Sort layers by AS$^l$ descending $\rightarrow L_{\text{text-aware}}$

**Stage 2: Enhance Vision in generation**
1: **For** each generation step $t$ **do**
2:   **if** layer $l \in L_{\text{text-aware}}$ **then**
3:     Enhance visual regions                       ▷ Equation (6)
4:   **else**
5:     Attention forward compute                    ▷ Equation (4)
6:   Compute multi - head attention outputs         ▷ Equation (7)
7:   Get probability of next token                  ▷ Equation (8)

---

## C.2 IMPLEMENTATION DETAILS OF REPETITION PHENOMENON

**Source of Text Perturbations.** We randomly sample 1,000 images from the COCO 2014 validation set (seed = 42). For each image, both LVLMs generate a caption using the prompt "Describe the image in detail." We collect all generated captions into a repository. To create a perturbed input for each image, we uniformly sample one caption from this repository, ensuring it does not correspond to the same image, and append it to the original prompt. All generations use greedy decoding with max_new_tokens = 512.

## C.3 IMPLEMENTATION DETAILS OF IDENTIFYING TEXT-AWARE LAYERS

In Equation (5), when calculating attention shifts, we select the last token generated by the model ($g$ = last token) and extract the corresponding attention weights. This design offers a key advantage: it enables convenient localization and extraction of attention weights regardless of the length of the generated text.

## C.4 THE ALGORITHM OF TCI

We show the two-stage algorithm of TCI in Table 6.

## C.5 DETAILS OF GPT-4O EVALUATION

Following prior work Huang et al. (2024); He et al. (2025), we employ GPT-4o to evaluate LVLMs' performance on LLaVA-Bench (In-the-Wild). The adapted prompt, derived from He et al. (2025), is presented in Table 7. All model responses were generated using greedy decoding with max_new_tokens = 512.

For each sample evaluation, GPT-4o was provided with the original image, the baseline model's response, and the TCI-augmented model's response. Three metrics were assessed: Accuracy: Measures alignment between the image and model output. GPT-4o assigns lower scores if inconsistencies (i.e., hallucinations) are identified. Detailedness: Reflects the comprehensiveness of the model's expressive capacity. Naturalness: Evaluates the fluency of generated text.

Table 7: The prompt used for GPT-4o evaluation.

---

**GPT-4o Prompt**

---

You are required to score the performance of two AI assistants in describing a given image. You should pay extra attention to the hallucination, which refers to the part of descriptions that are inconsistent with the image content, such as claiming the existence of something not present in the image or describing incorrectly in terms of the counts, positions, or colors of objects in the image.

Please rate the responses of the assistants on a scale of 1 to 10, where a higher score indicates better performance, according to the following criteria:

1: Accuracy: whether the response is accurate with respect to the image content. Responses with fewer hallucinations should be given higher scores.

2: Detailedness: whether the response is rich in necessary details. Note that hallucinated descriptions should not count as necessary details.

3: Naturalness: assess the language quality, focusing on: fluency of sentence structure, appropriateness of word choice, smoothness of language flow, absence of awkard or unnatural phrasing.

Please output the scores for each criterion, containing only two values indicating the scores for Assistant 1 and 2, respectively. The two scores are separated by a space. Following the scores, please provide an explanation of your evaluation, avoiding any potential bias and ensuring that the order in which the responses were presented does not affect your judgment.

{Question}
{ }
{End of Question}

{Assistant 1}
{ }
{End of Assistant 1}

{Assistant 2}
{ }
{End of Assistant 2}

Output format:
Accuracy:
Reason:
Detailedness:
Reason:
Naturalness:
Reason:

---

# D  ADDITIONAL EXPERIMENTS RESULTS

## D.1  HEAT MAP OF LAYER-WISE ATTENTION SHIFT

Figure 7 visualizes the layer-wise shifts in attention toward visual regions before and after text perturbation. The two models exhibit distinct shift patterns : the top 5 layers with the largest shifts for LLaVA-1.5-7B are $\{0, 1, 14, 15, 17\}$, whereas for Qwen-VL-Chat-7B they are $\{0, 1, 17, 20, 31\}$.

## D.2  RESULTS OF POPE

Table 8 shows the results of three splits (random, popular, and adversarial) of POPE. Our method consistently outperforms previous approaches on multiple subsets, demonstrating its effectiveness as a training-free strategy across different performance levels.

# E  CASE STUDY

Figure 8 and 9 presents several illustrative cases demonstrating the effectiveness of TCI in reducing hallucinations. Without TCI, the model generates descriptions inconsistent with the image (highlighted in bold red), such as references to "people" and "chairs". In contrast, TCI not only mitigates such hallucinations but also preserves critical image details.



(a) Layer-wise attention shift heat map of LLaVA-1.5-7B



(b) Layer-wise attention shift heat map of Qwen-VL-Chat-7B

Figure 7: Layer-wise attention shift heat maps of models

Table 8: **Accuracy and F1 scores on POPE** popular, adversarial, and random splits, with best in **bold** and second-best underlined.

| Split | Method | LLaVA-1.5-7B | | Qwen-VL-Chat-7B | |
|---|---|---|---|---|---|
| | | **Accuracy** | **F1 Score** | **Accuracy** | **F1 Score** |
| Random | Direct Sample | 83.29 | 81.33 | 84.37 | 82.67 |
| | Greedy | 89.60 | **89.72** | 88.83 | 87.72 |
| | VCD | 88.07 | 88.30 | 87.90 | 86.78 |
| | OPERA | 89.20 | 88.81 | 88.76 | 87.84 |
| | **TCI** | **90.13** | **89.72** | **89.40** | **88.47** |
| Popular | Direct Sample | 81.88 | 80.06 | 84.13 | 82.06 |
| | Greedy | 86.20 | 86.80 | 87.20 | 86.17 |
| | VCD | 84.40 | 85.26 | 85.40 | 84.31 |
| | OPERA | 86.64 | 86.62 | 87.44 | 86.68 |
| | **TCI** | **88.67** | **88.36** | **87.97** | **87.11** |
| Adversarial | Direct Sample | 78.96 | 77.57 | 82.26 | 80.37 |
| | Greedy | 79.77 | 81.78 | 84.40 | 83.64 |
| | VCD | 78.63 | 80.89 | 82.47 | 81.95 |
| | OPERA | 81.24 | 81.38 | 84.78 | 83.45 |
| | **TCI** | **83.77** | **84.10** | **85.00** | **84.43** |

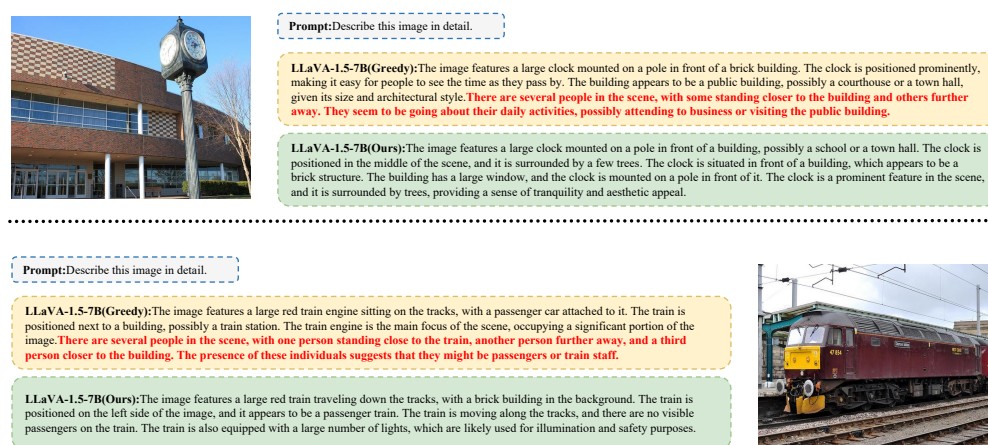

Figure 8: Examples of TCI on LLaVA-1.5-7B. Hallucinatory content is highlighted in bold red.

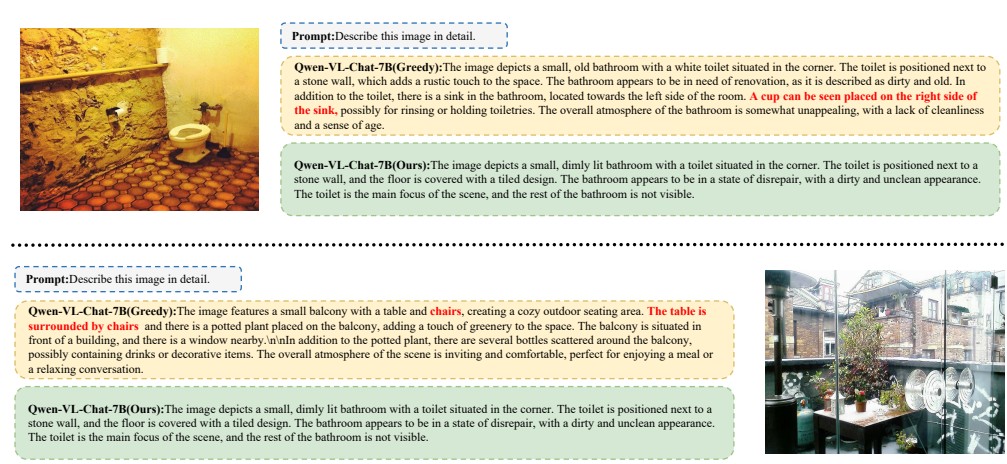

Figure 9: Examples of TCI on Qwen-VL-Chat-7B. Hallucinatory content is highlighted in bold red.

