# OpenReview forum: "TCI: Mitigating Hallucination in LVLMs Via Text Contrastive Intervention"
_ICLR.cc/2026/Conference — ICLR 2026 Conference Withdrawn Submission_

### Official Review · Reviewer_FyVC · 2025-10-27

**Soundness:** 2
**Presentation:** 2
**Contribution:** 2
**Rating:** 2
**Confidence:** 4

**Summary:**

This paper introduces Text Contrastive Intervention (TCI), a training-free method designed to reduce hallucinations in Large Vision-Language Models (LVLMs). Inspired by the observed repetition phenomenon—where models tend to verbatim repeat input text in the face of conflicting image-text cues—TCI works by perturbing the textual input to identify attention layers most susceptible to language bias, then amplifies visual attention in those layers during inference. The approach is benchmarked against state-of-the-art strategies on POPE and CHAIR datasets using LLaVA-1.5-7B and Qwen-VL-Chat-7B models, showing notable improvements in hallucination mitigation metrics and competitive performance on other quality benchmarks.

**Strengths:**

**Conceptual Shift**: The paper takes a motivated, empirically-anchored pivot from the dominant image-based perturbation paradigm for contrastive hallucination mitigation, providing a systematic text-centric approach. This focus on text perturbation is well supported by qualitative and quantitative explorations of the “repetition phenomenon” and may help broaden the intervention toolkit for LVLM hallucination.

**Clarity and Transparency**: The methodology is clearly documented including mathematical formulas (e.g., Equation defining attention shifts, enhancement), algorithmic steps (Table 6), and an explicit ablation on hyperparameter $\alpha$ and layer selection $k$.

**Empirical Rigor**: Results on standard hallucination and captioning datasets (Table 2 and Table 3) are consistently strong, demonstrating TCI's advantage over prior decoding-based methods (VCD, OPERA) across multiple models and settings. Ablations help validate and contextualize the key components.

**Weaknesses:**

**Critical Concerns Regarding Novelty**: The paper’s claims of novelty seem to be overstated. The diagnostic component—perturbing input text to surface hallucinations has strong precedents. Language-Contrastive Decoding (LCD; Manevich & Tsarfaty, 2024) directly targets hallucinations via contrastive language manipulations, yet is omitted from both discussion and benchmarking. Also,  the "repetition phenomenon" that motivates TCI is a direct manifestation of the same underlying principle that ICD and other text-debiased decoding approaches.

The intervention mechanism has also been explored by recent advances. TCI globally amplifies visual attention at the layer level, whereas methods like VisFlow perform more precise, head-level attention re-weighting. Additionally, works such as Devils in Middle Layers highlight specific middle-layer roles and introduce more structured visual-semantic integration strategies.

Given this, in my view, the paper's main novelty lies not in its diagnostic or interventional components individually, but in its hybrid nature. TCI effectively bridges two distinct families of training-free methods: it uses a diagnostic technique from the contrastive decoding family to guide an intervention from the attention manipulation family. But this unique positioning is not articulated or evaluated.

**One-off calibration**: Layer selection for intervention is determined via a one-off calibration using 1K COCO images. The paper provides no analysis of how stable this identified set of layers is. It is an open question whether a different random sample of images, or a dataset from a different domain (e.g., medical imaging), would yield the same set of layers.

**Hyperparameter Sensitivity**: The amplification factor α requires extensive per-task tuning. The reported optimal values diverge not just across models but within the same model across benchmarks. This undermines the claim of a lightweight “plug-and-play” method since each new deployment demands calibration and sweeping of α.

**Limited & Outdated Model Coverage**: Only small (7B-scale) and older VLLMs are tested. The paper neither examines larger or newer architectures nor evaluates whether the motivating “repetition phenomenon” persists in state-of-the-art models (e.g., Qwen3). Generality remains speculative.

**Baseline**: Important baselines such as LCD are missing. This weakens the completeness of empirical claims.

**Questions:**

On Novelty: Could the authors clarify the conceptual distinction between TCI’s diagnostic stage and prior contrastive decoding techniques such as LCD and ICD? In particular, what new insight does TCI introduce beyond existing perturbation-based hallucination detection?

On Intervention Granularity: TCI uniformly amplifies all attention heads in selected layers, whereas recent methods (e.g., VisFlow) operate at the head level. Why is a coarse, layer-level adjustment sufficient, and how does it compare to finer-grained alternatives?

On Robustness of Layer Selection: How stable is the set of top-k “text-aware” layers identified during calibration? Would different calibration subsets, domains, or model architectures produce substantially different layer selections?

On Practicality and Hyperparameter Sensitivity: The amplification factor α varies substantially across models and benchmarks. How do the authors reconcile this tuning burden with the claim of a model-agnostic, plug-and-play method? Is there a principled way to choose α without exhaustive sweeping?

On Model Scale and Modernity: Given that only 7B-scale models are evaluated, can the authors comment on whether the observed repetition phenomenon persists in larger or newer VLLMs (e.g., Qwen3)? Are the current findings expected to generalize?

---

### Official Review · Reviewer_2BRG · 2025-10-28

**Soundness:** 2
**Presentation:** 3
**Contribution:** 2
**Rating:** 4
**Confidence:** 4

**Summary:**

This paper proposes **Text Contrastive Intervention (TCI)**, a training-free method to mitigate hallucinations in Large Vision-Language Models (LVLMs). Instead of perturbing the visual modality as in prior contrastive decoding approaches, TCI perturbs the text modality to amplify language priors and identify attention layers most susceptible to text bias. By enhancing visual attention in these layers during the forward pass, TCI reduces over-reliance on language priors. Experiments on benchmarks such as POPE, CHAIR, and LLaVA-Bench demonstrate consistent improvements over baselines like VCD and OPERA, validating the method’s efficiency and effectiveness.

**Strengths:**

1.Originality: The idea of text-based contrastive intervention is novel and intuitive. Prior work mainly perturbs the visual input, while this work innovatively targets text to directly expose and counteract language priors—a fresh angle in hallucination mitigation.

2.Quality: The paper provides clear motivation, well-defined methodology, and comprehensive experiments on multiple LVLMs and benchmarks. The ablation studies (on α and top-k layers) convincingly support the design choices.

3.Significance: The method is lightweight and model-agnostic, requiring no additional training, which makes it practical for real-world LVLM deployment. Its consistent performance gains over strong baselines highlight potential impact.

**Weaknesses:**

1.Limited scope of intervention: TCI focuses solely on attention layers, leaving out other components (e.g., FFNs), which might also contribute to hallucination. This limits the completeness of its intervention strategy.

2.Empirical depth: Although experiments cover key benchmarks, they are mostly limited to two models (LLaVA-1.5-7B, Qwen-VL-Chat-7B). Results on larger-scale or instruction-tuned LVLMs (e.g., GPT-4V, Gemini) would strengthen generalization claims.

3.Analysis depth: While the “repetition phenomenon” is an interesting observation, its theoretical underpinning is shallow; the paper mainly provides empirical evidence without deeper causal analysis.

**Questions:**

1.How sensitive is TCI to the choice of text perturbation strategy? Would semantically different perturbations (e.g., antonym replacements) yield similar results?

2.Have you tested whether TCI introduces new types of errors (e.g., overemphasis on irrelevant visual details)?

3.Since TCI modifies attention activations during inference, could it conflict with other decoding-time interventions (e.g., self-reflection or CoT prompting)?

---

### Official Review · Reviewer_Um31 · 2025-10-31

**Soundness:** 2
**Presentation:** 3
**Contribution:** 2
**Rating:** 2
**Confidence:** 4

**Summary:**

This paper introduces Text Contrastive Intervention (TCI), a training-free approach to mitigate hallucinations in Large Vision-Language Models (LVLMs). Instead of conventional contrastive methods that perturb the visual modality, TCI perturbs the text input to directly amplify language priors, then identifies attention layers most susceptible to text bias and selectively boosts visual-attention weights during generation. The main rationale is built on the "repetition phenomenon," where LVLMs tend to repeat input text when facing conflicts between visual and linguistic signals, indicating language bias as a root cause of hallucination. Extensive experiments on several benchmarks demonstrate that TCI reduces hallucinations without requiring retraining and with minimal data.

**Strengths:**

1. TCI is a training-free method that can be applied post hoc to existing LVLMs during inference, increasing its practical value and ease of adoption. The approach relies on identifying text-biased layers via attention shift dynamics, and then only intervenes there.

2. The paper systematically evaluates TCI across standard metrics (POPE, CHAIR, LLaVA-Bench), LVLMs (LLaVA, Qwen), and various experimental settings. Quantitative gains are well-documented.

3. The paper commits to releasing code, details hyperparameters, and provides additional implementation specifics in appendices, facilitating reproducibility.

**Weaknesses:**

1. The core of the TCI method is built on the observed repetition behavior (repetition phenomenon) and assumes it stems from language priors. However, the paper does not delve deeply into why the model tends to "repeat" the input text rather than generating "creative" or "speculative" alternative content. This behavior may only be an extreme manifestation of language priors, rather than a universal mechanism driving all types of hallucinations. Furthermore, the "repetition phenomenon" is induced by artificially constructed text perturbations (such as appending unrelated image descriptions to the current prompt). Moreover, in real, unperturbed user input scenarios, the phenomenon of the model directly "verbatim repeating" the input text may not be common, limiting the relevance and universality of this observation in practical applications.

2. There is a lack of deeper architectural explanation for why the top k text-biased layers, as measured by $AS^l$ (attention shift), always correspond to specific indices. The paper observes significant differences in the attention shift patterns between the two models; for example, Qwen-VL-Chat-7B's top-5 layers {0, 1, 17, 20, 31} are not primarily concentrated in shallow (early layers) and middle (intermediate layers) like LLaVA-1.5-7B's {0, 1, 14, 15, 17}, but are more dispersed, including deep layers (such as the last layer 31). Although the paper mentions the differences in projection mechanisms between the two models in the experimental setup, it does not deeply explain why Qwen exhibits deep layer shifts.

3. Although the paper cites and evaluates contrastive decoding methods like VCD and OPERA as baselines, the discussion and quantitative comparison with the most language-contrastive intervention measures (such as Language-Contrastive Decoding (LCD) (Manevich & Tsarfaty, 2024)) and other decoding-side mitigation methods (such as SID, DeGF, DeCo, AGLA...) are insufficient. These omissions may lead to an inappropriate evaluation of TCI's novelty, as similar methods (e.g., LCD reduces language bias by contrasting LVLM and LLM outputs) are conceptually similar but not included in the benchmark evaluation.

4. It is unclear how robust the layer selection or parameter α is for other LVLM architectures or prompts, and there is a lack of sensitivity analysis for tuning α and k across prompts and domains. There is a risk that the method is over-tuned to the target models and data, and its generalization claims may be exaggerated. Deployment in practice would benefit from guidelines on robust default values, sensitivity analysis under domain shifts, and failure detection, while the current analysis is limited to one-time scans per dataset/model.

**Questions:**

1. Could you expand the comparisons to include recent SOTA methods for hallucination mitigation in LVLMs, such as SID (Self-Introspective Decoding), DeGF (Self-Correcting Decoding with Generative Feedback), DeCo (Decoding by Contrasting Layers), and AGLA (Assembly of Global and Local Attention)? Providing both qualitative discussions and quantitative benchmarks against these would better highlight AGE's novelty and relative performance?

2. The paper hypothesizes that the repetition behavior arises from language priors, but could the authors elaborate on potential alternative explanations (e.g., why repetition occurs instead of more "creative" hallucinations like fabricating non-existent objects)?

3. Given the observed differences in top-k layers between LLaVA-1.5-7B (concentrated in shallow/middle layers) and Qwen-VL-Chat-7B (more dispersed, including deep layers like 31), can the authors provide a deeper analysis of how the models' projection mechanisms (MLP vs. cross-attention) or other architectural factors contribute to these patterns?

---

### Official Review · Reviewer_rxYj · 2025-11-01

**Soundness:** 3
**Presentation:** 3
**Contribution:** 2
**Rating:** 4
**Confidence:** 4

**Summary:**

This paper proposes a simple mechanism that strengthens the attention weights of visual tokens, which the authors claim helps reduce hallucination.

**Strengths:**

The proposed method is simple and easy to implement. If it also proves effective in real-world scenarios, it has strong potential for practical application.

**Weaknesses:**

1. The paper claims that the proposed approach is inspired by the “repetition phenomenon.” However, after carefully reading the methodology section, I could not identify any clear connection between the proposed method and the repetition phenomenon.

2. The description of the method in Section 3.2 reads naturally, whereas Section 3.1 appears less natural. I have some concerns regarding this layer selection process:
   (a) Could the authors clarify whether the data used for selecting layers also participates in the evaluation described in the experimental parts?
   (b) Empirically, adjacent layers tend to exhibit similar behavior. Therefore, in Table 5, the baseline should ideally sample layers more evenly to ensure broader coverage. For instance, if the model has 32 layers, the baseline could select layers 0, 6, 12, 18, and 24.
   (c) The authors state that, for example, the top five layers with the largest shifts for LLaVA-1.5-7B are {0, 1, 14, 15, 17}, while for Qwen-VL-Chat-7B they are {0, 1, 17, 20, 31}. However, the layer selection procedure described in Eq. (2) involves randomness. The paper lacks an analysis of how this randomness affects the final layer selection and overall performance. For instance, could rerunning the full experiment with a different random seed lead to significantly different layer selections, and hence, a different final performance?

3. The experiments do not report the length of the output tokens. Comparing hallucination levels is only meaningful when the lengths of the generated outputs are similar.

**Questions:**

see Weaknesses.

---

### Note · Authors · 2025-11-17

**Comment:**

We appreciate the comments made by all the reviewers and have decided to withdraw and revise the manuscript.Thank you again.

**Withdrawal Confirmation:**

I have read and agree with the venue's withdrawal policy on behalf of myself and my co-authors.